# Five Bonds to Carbon through Tri-Coordination in $Al_3C_3^{-/0}$

Abdul Hamid Malhan [1] , Venkatesan S. Thimmakondu [2],* and Krishnan Thirumoorthy [1],*

1     Department of Chemistry, School of Advanced Sciences, Vellore Institute of Technology, Vellore 632 014, Tamil Nadu, India; abdulhamid.malhanb2020@vitstudent.ac.in

2     Department of Chemistry and Biochemistry, San Diego State University, San Diego, CA 92182-1030, USA

*     Correspondence: vthimmakondusamy@sdsu.edu (V.S.T.); thirumoorthy.krishnan@vit.ac.in (K.T.)

**Abstract:** Here, five bonds to carbon through tri-coordination are theoretically established in the global minimum energy isomers of $Al_3C_3^-$ anion (**1a**) and $Al_3C_3$ neutral (**1n**) for the first time. Various isomers of $Al_3C_3^{-/0}$ are theoretically identified using density functional theory at the PBE0-D3/def2-TZVP level. Chemical bonding features are thoroughly analyzed for these two isomers (**1a** and **1n**) with different bonding and topological quantum chemical tools, such as adaptive natural density partitioning (AdNDP), Wiberg Bond Indices (WBIs), nucleus-independent chemical shifts (NICS), and atoms in molecules (AIM) analyses. The structure of isomer **1a** is planar with $C_{2v}$ symmetry, whereas its neutral counterpart **1n** is non-planar with $C_2$ symmetry, in which its terminal aluminum atoms are out of the plane. The central allenic carbon atom of isomers **1a** and **1n** exhibits tri-coordination and thus makes it a case of five bonds to carbon, which is confirmed through their total bond order as observed in WBI. Both the isomers show σ- and π-aromaticity and are predicted with the NICS and AdNDP analyses. Further, the results of ab initio molecular dynamics simulations reveal their kinetic stability at room temperature; thus, they are experimentally viable systems.

**Keywords:** $Al_3C_3^{-/0}$; five bonds to carbon; anti-van't Hoff-Le Bel; σ-aromaticity; π-aromaticity; bonding; computational chemistry





## 1. Introduction

The concept of five bonds to carbon became indispensable since the discovery of methanium ion ($CH_5^+$) in the laboratory in 1950 [1]. Recording the infrared spectra of this simple protonated methane molecule was quite challenging, as it took almost five decades from its discovery [2]. The theoretical investigation of lithium carbides such as $CLi_5$ and $CLi_6$ [3] and the experimental realization of $CLi_6$ through mass spectroscopic measurements further motivated the interest in hyper-coordinate carbon molecules [4]. While computational studies on $Si_2(CH_3)_7^+$ [5] and $C(CH_3)_5^+$ [6] provided further guidance on hyper-coordinate behavior of group 14 elements, it is the experimental observations such as $[CCH_3]_6^{2+}$, $HC[Au(PPh_3)]_4^+$, $[(C_6H_5)_3PAu_5C]^+$, $[(Ph_3PAu)_6C]^{2+}$, $C_6[CH_3]_6^{2+}$, etc., that gave chemists the real grandeur of hyper-coordinate carbon molecules [7–11]. Akiba and co-workers have shown penta- and hexa-coordinate anthracene moieties through x-ray crystallography and ab initio calculations [12,13]. The iron–molybdenum nitrogenase cofactor existing in diazotrophs is a clear example of hexa-coordinate carbon in biological systems [14]. From the well-known concept of molecules with a planar tetra-coordinate carbon (ptC) atom [15–30], the idea was extended to planar penta-coordinate carbon (ppC) [31–40] and planar hexa-coordinate carbon (phC) [41–44]. Hill and coworkers experimentally reported the existence of a penta-coordinate carbon atom in 1981 [45]. The penta-coordinate carbon atom was also theoretically reported by Gleiter and coworkers [46] in the $Cp_2Zr[CH_2(BH\{C_6F_5\}_2)_2]$ complex. In 1996, the experimental proof of the first complex with a hyper-coordinate ylidic carbon atom was also reported by Jones and coworkers [47]. While a gradual amount of progress has been made in these classes of molecules to date, to a larger extent in the literature, the concept of making hyper-coordinate carbon molecules was predominantly

focused on making single bonds to the central carbon atom irrespective of whether they are planar or non-planar [48]. However, in this article, the intent is to make five bonds to a carbon atom through tri-coordination instead of penta-coordination, as is normally carried out. To this end, we have theoretically investigated the aluminum–carbon cluster, $Al_3C_3^{-/0}$, in both its anion and neutral forms and established the fact that the global minimum isomers (**1a** and **1n**) contain five bonds to carbon through tri-coordination (see Figures 1 and 2).

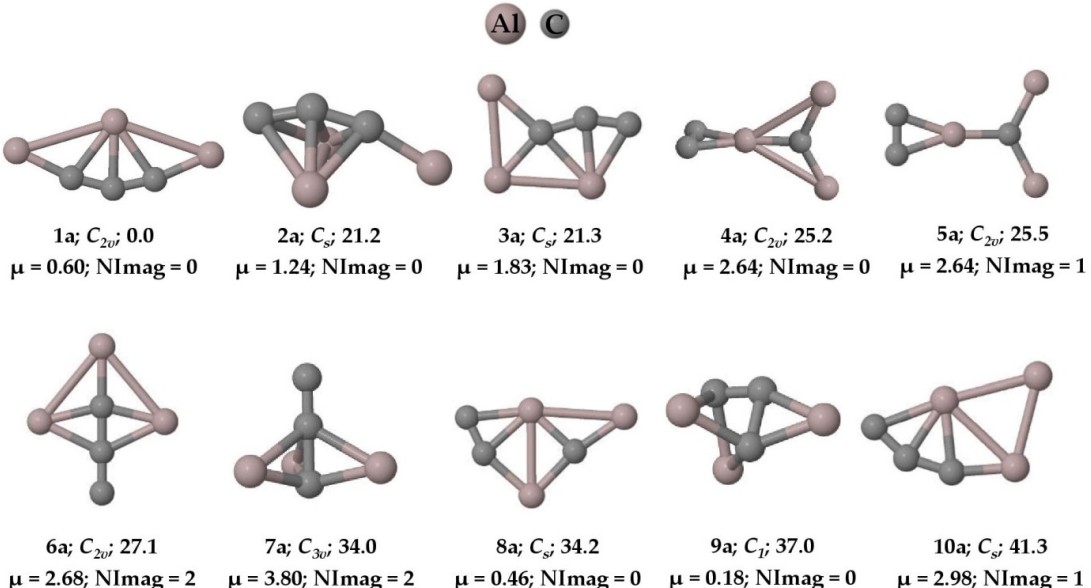

**Figure 1.** Ten low-lying isomers of $Al_3C_3^-$ with the ZPVE-corrected relative energies (in kcal mol$^{-1}$), dipole moments (in Debye), and the number of imaginary frequencies (NImag) obtained at the PBE0-D3/def2-TZVP level.

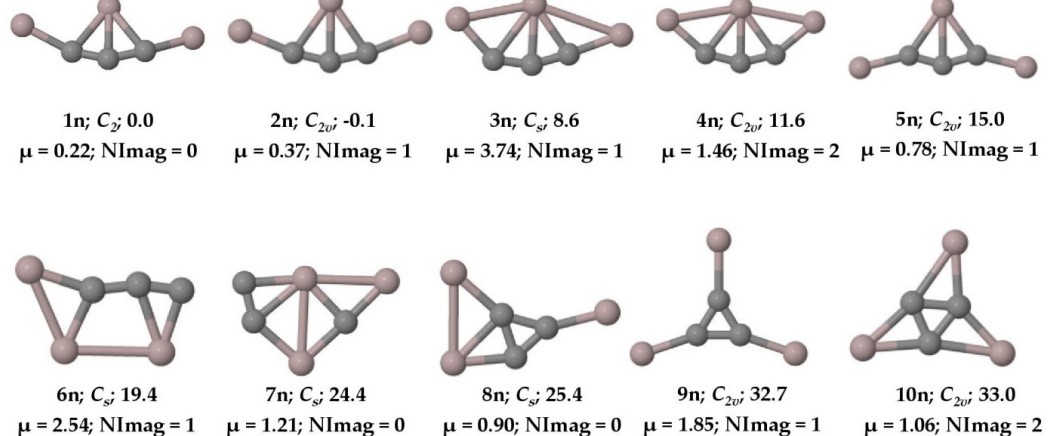

**Figure 2.** Ten low-lying isomers of $Al_3C_3$ with ZPVE-corrected relative energies (in kcal mol$^{-1}$), dipole moments (in Debye), and the number of imaginary frequencies (NImag) obtained at the PBE0-D3/def2-TZVP level.

Various aluminum–carbon clusters have continuously been investigated [16,17,49–52] as they have potential applications in energy storage [53] and the production of nanopowders [54,55] and solar cells [56]. Zheng and coworkers theoretically and experimentally explored the $Al_4C_6^{-/0}$ system and found that the global minimum isomer in the neutral state contains the planar hexa-coordinate aluminum [57]. In 2022, Kalita et al. reported the planar penta-coordinate Al and Ga centers in $Cu_5Al_2^+$ and $Cu_5Ga_2^+$ systems, which were global minimum structures, and found that the stabilizing factor was σ-aromaticity [58].

Recently, Malhan et al. reported the $Al_2C_4H_2$ system with ptC, planar tetra-coordinate aluminum (ptAl), and planar penta-coordinate aluminum (ppAl) atoms with aromatic characteristics [59]. The aforementioned discoveries have also inspired researchers to look for further systems containing hyper-coordinate main group elements, such as group 13 elements. Wang and coworkers reported $B_8^-$ and $B_9^-$ clusters exhibiting planar hepta- and octa-coordinate central boron atoms with combined experimental and computational studies [60]. Li et al. reported the global minimum structure of the $BCu_5H_5^-$ system with planar penta-coordinate boron (ppB) [61]. The global minimum isomer containing the ppB atom in the $B_6H_5^+$ system with aromatic characteristics was reported by Wu and coworkers. In 2021, Khatun et al. reported $BAl_4Mg^{-/0/+}$ [62], in which the global minimum structures were found to have a planar tetra-coordinate boron (ptB) atom in its anionic and cationic forms as well as a ppB atom in the neutral state. In 2022, Das and coworkers explored the potential energy surface of $CB_6Al^{0/+}$ [63] and found that both neutral and cation contain planar hexa-coordinate boron (phB) atoms in their global minima. Thompson et al. experimentally reported the ptAl species [64]. The ptAl species of calix [4] pyrrole aluminate was also experimentally reported by Greb and coworkers in 2019 [65]. In 2023, Merino and coworkers also reported a quasi-ptC atom in the $CAl_{11}^-$ system [66]. These distinctive bonding arrangements demonstrate not only the fundamental importance of improving our knowledge of chemical bonding but also a completely new class of molecules in the world of chemistry. Herein, the present work reports the $Al_3C_3^{-/0}$ system with tri-coordination; that is, five bonds to the central allenic carbon atom via computational quantum chemical modeling. The aluminum and carbon-based molecules have potential applications ranging from cluster assembled materials [67,68], energy storage [69], and two-dimensional donor materials in solar cells [70]. Dong et al. [71] have already reported the isomer **2n** of the $Al_3C_3$ system with hydrogen storage properties both experimentally and theoretically at MP2/6-311+G* level of theory, which gives us confidence that the $Al_3C_3^{-/0}$ system investigated here, has a high chance of synthetic viability in the future.

## 2. Computational Methodology

The initial geometries of the $Al_3C_3^{-/0}$ system were first generated through chemical intuition, and then, using the in-house Python code, all other possible geometries were explored on these two potential energy surfaces (PESs) using density functional theory (DFT). All geometries were fully optimized using the hybrid functional PBE0 [72] coupled with Grimme's dispersion correction (D3) [73,74] and the def2-TZVP [75,76] basis set. Frequency calculations were carried out at the same level of theory to ensure whether the optimized geometries are true minima or maxima or higher-order saddle points. To get more characteristic features on the chemical bonding of isomers **1a** and **1n,** the natural bond order (NBO) analysis [77], adaptive natural density partitioning (AdNDP) analysis [78,79], and Wiberg bond indices (WBIs) [80] were performed at the PBE0-D3/def2-TZVP level. The nucleus-independent chemical shift (NICS) [81] calculations for **1a** and **1n** structures were carried out at the same level to analyze the aromatic behavior of these systems. Atoms in molecules (AIM) analysis [82] of the Laplacian of electron density and electron localization function (ELF) [83] were carried out for isomers **1a** and **1n** using the wave function file generated by the Gaussian program [84] at the PBE0-D3/def2-TZVP level. The dynamic stabilities of **1a** and **1n** were evaluated using the atom-centered density matrix propagation (ADMP) [85] at the same level of theory. The AdNDP and ELF were analyzed through the Multiwfn program [86]. All the calculations were performed using the Gaussian 16 package [84].

## 3. Results and Discussion

The PESs of the $Al_3C_3^{-/0}$ are explored, and we found that the global minimum energy geometry (**1a** and **1n**) of both the anion and neutral system contains a carbon atom (C4) with two π bonds and three σ bonds, exhibiting a total of five bonds through tri-coordination. The ten low-lying isomers of the anion and the neutral system are given in

Figures 1 and 2, respectively. All other isomers on the PES of the $Al_3C_3^{-/0}$ system are given in the Supplementary Materials in Figures S1 and S2, respectively. The $Al_3C_3^-$ system has a singlet spin state and the $Al_3C_3$ system corresponds to a doublet. Isomer **1a** is a planar structure with $C_{2v}$ symmetry, whereas isomer **1n** shows $C_2$ symmetry, in which the terminal aluminum atoms are out of the plane (see Figure 3). The $Al_3C_3^{-/0}$ system also has structures that exhibit planar penta-coordinate aluminum (ppAl) and planar tetra-coordinate carbon (ptC) atoms as local minimum energy isomers (**8a**, **7n**; and **3a**, **13a**, and **8n**, respectively) on their PESs. Nevertheless, our focus here is on the global minimum energy isomers, **1a** and **1n,** which exhibit five bonds to carbon through tri-coordination.

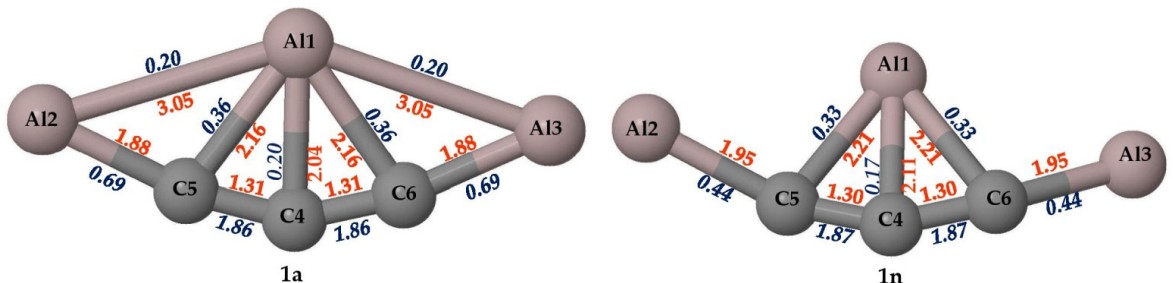

**Figure 3.** The bond lengths (in Å, red) and the Wiberg bond orders (in blue) are obtained at the PBE0-D3/def2-TZVP level for isomers **1a** and **1n**.

## 3.1. Wiberg Bond Indices

The WBI values obtained from NBO analysis for the allenic carbon (C4) atom in isomers **1a** and **1n** are critically analyzed. The WBI values and the bond distances are given in Figure 3. The standard covalent bond lengths of C–Al and C=C are 2.01 and 1.34 Å, respectively, which are in close agreement with the obtained values. The C4–Al1 bond length of isomer **1n** having 2.11 Å is slightly higher than that of isomer **1a** with 2.04 Å. The WBI values for the C=C bond in isomers **1a** and **1n** are 1.86 and 1.87, respectively, which confirms the presence of π bonds in both isomers. This further proves that in both cases, the C4 atom makes two π bonds with neighboring atoms. The C4–Al1 bond with WBI values of 0.20 and 0.17 in isomers **1a** and **1n** suggests the dative bonding nature of the fifth bond. The total bond order of C4 in isomers **1a** and **1n** is 3.99 and 3.97, respectively. This indicates that the allenic carbon atom, C4, is surrounded by eight electrons but still makes two π bonds with neighboring carbon atoms and also forms an additional dative bond with the central aluminum atom.

## 3.2. Adaptive Natural Density Partitioning (AdNDP) Analysis

To further analyze the bonding scenario, the AdNDP analysis was carried out for the delineation of n-center 2-electron (nc-2e) bonds in the investigated systems. The generated AdNDP orbitals with occupation numbers (ON) for isomers **1a** and **1n** are shown in Figure 4. As the neutral system is in a doublet state, only alpha orbitals are considered for this analysis. The tri-coordinated C4 atom has two 2c–2e σ and two 2c–2e π bonds with its neighboring carbon atoms, with ON 1.99 |e| and 1.85 |e|, respectively, which confirms the presence of alternating π bonds in the isomer **1a**. It also exhibits delocalization of electron densities through 3c–2e σ, 4c–2e σ, 3c–2e π, and 4c–2e π bonds with ON ranging from 1.89 |e| to 2.00 |e, which support the tri-coordination in the structure. The two 2c–2e π bonds in isomer **1n** confirm the presence of alternating π bonds which also exhibit delocalization of electron densities through multi-center 2e σ and π bonds, which supports the structural stability. To support the observed AdNDP bonding pattern, the nucleus-independent chemical shift (NICS) values are also calculated for isomers **1a** and **1n**, which are shown in Figure S3. The negative values of NICS (0) and NICS (1) also confirm the presence σ- and π-aromatic nature in both the isomers, respectively.

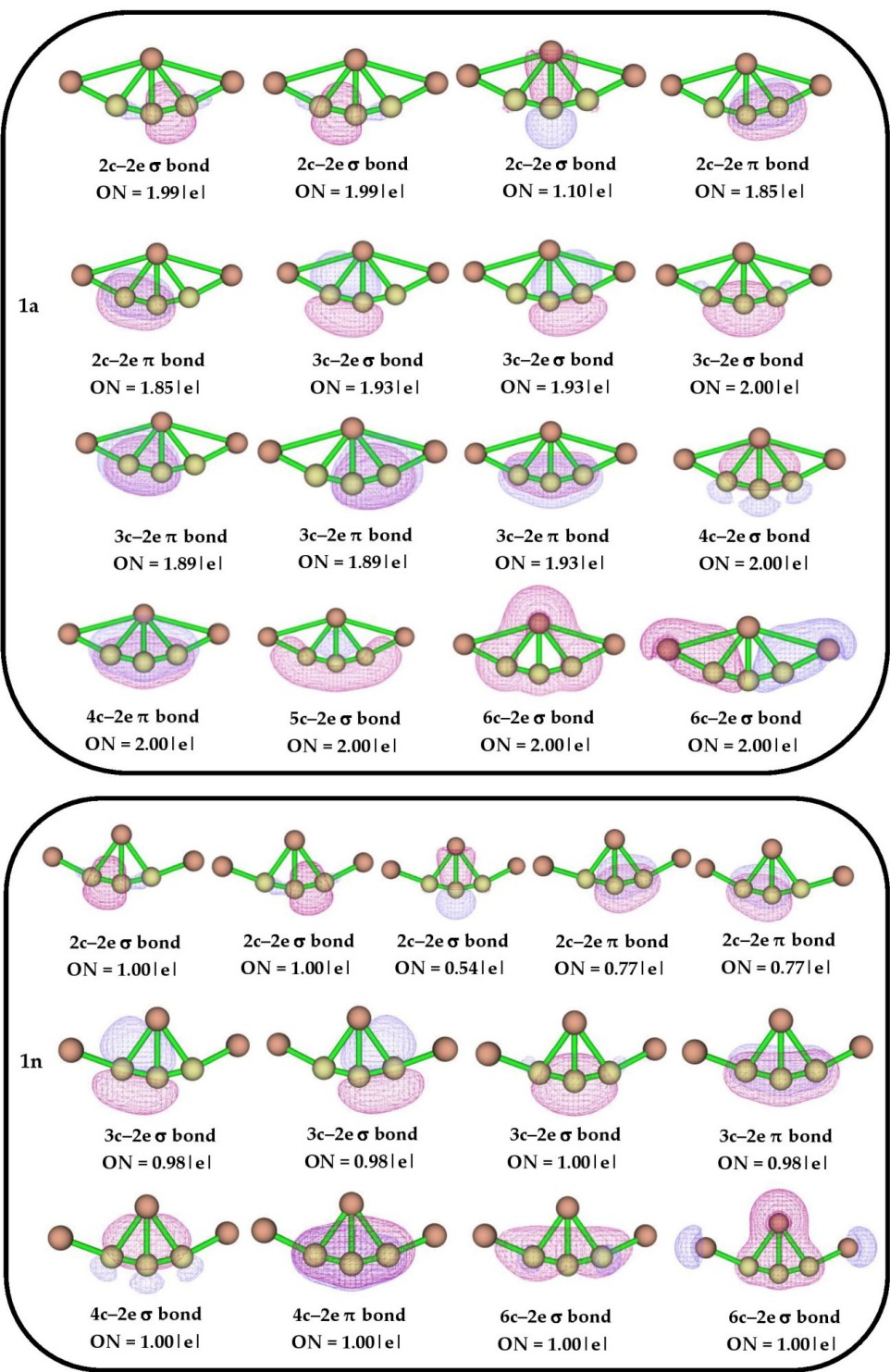

**Figure 4.** AdNDP bonding patterns with occupation numbers (ONs) for isomers **1a** and **1n**.

### 3.3. Atoms in Molecule (AIM) Analysis

The AIM analysis is carried out to gain insight into the bonding characteristic features. The color-filled plots of the electron localization function (ELF) and contour diagram for the Laplacian of electron density ($\Delta^2\rho(r)$) for isomers **1a** and **1n** are shown in Figure 5. The ELF plot of isomers **1a** indicates the interaction of the C4 atom with its neighboring atoms and supports the delocalization of electron densities within the molecule. Apart from the terminal Al2 and Al3 atoms, which are out of the plane, isomer **1n** also supports the delocalization of electron densities around the C4 atom. In **1a**, bond critical points (BCPs)

between the C4 and its neighboring atoms support the existence of bond paths. Isomer **1n** has BCPs between C4 and its adjacent carbon atoms and also has a ring critical point (RCP) which dictates the dominant aromatic characteristics in the structure.

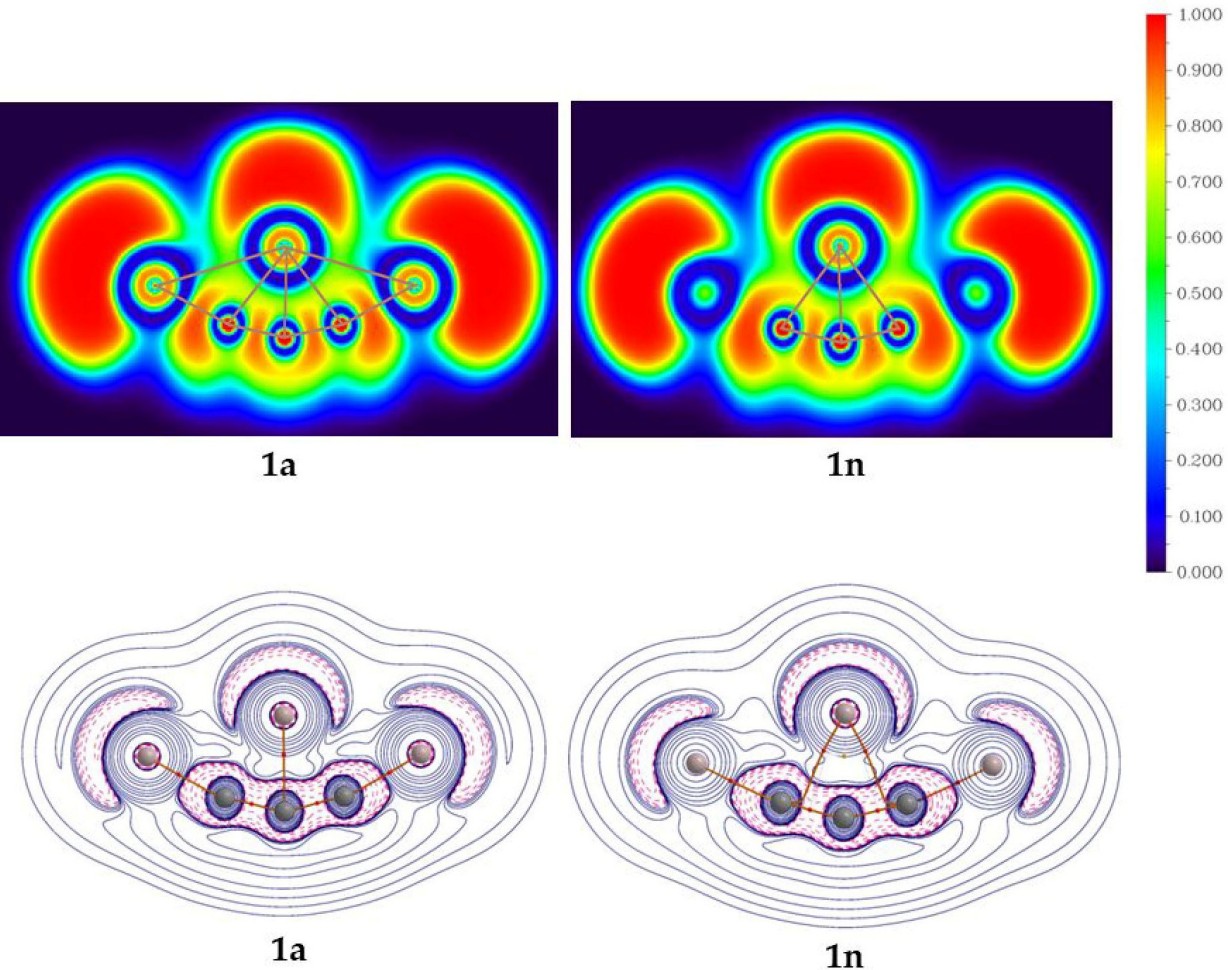

**Figure 5.** The color-filled map of ELF (top row) and the Laplacian of electron density ($\Delta^2\rho(r)$) with bond paths (bottom row) for isomers **1a** and **1n** at the PBE0-D3/def2-TZVP level.

### 3.4. Kinetic Stability

The ab initio molecular dynamics (AIMD) simulations are carried out for 3000 fs at 298 K and 1 atm pressure using the ADMP approach to explore the kinetic stability of the investigated structures. The time evolution of energy plots for isomers **1a** and **1n** are given in Figure 6. Slight structural deformation occurs during the simulation, which causes an increase in nuclear kinetic energy. However, the present data reveal that the overall geometry is not completely destroyed, which indicates that these molecules are kinetically stable apart from their thermodynamic stability. As expected, for both isomers **1a** and **1n**, the five bonds to the C4 atom remain the same throughout the simulation period. The structural stability of these isomers is well maintained during the simulation, and no isomerization or other structural modifications occur in these molecules, suggesting that they are indeed kinetically stable.

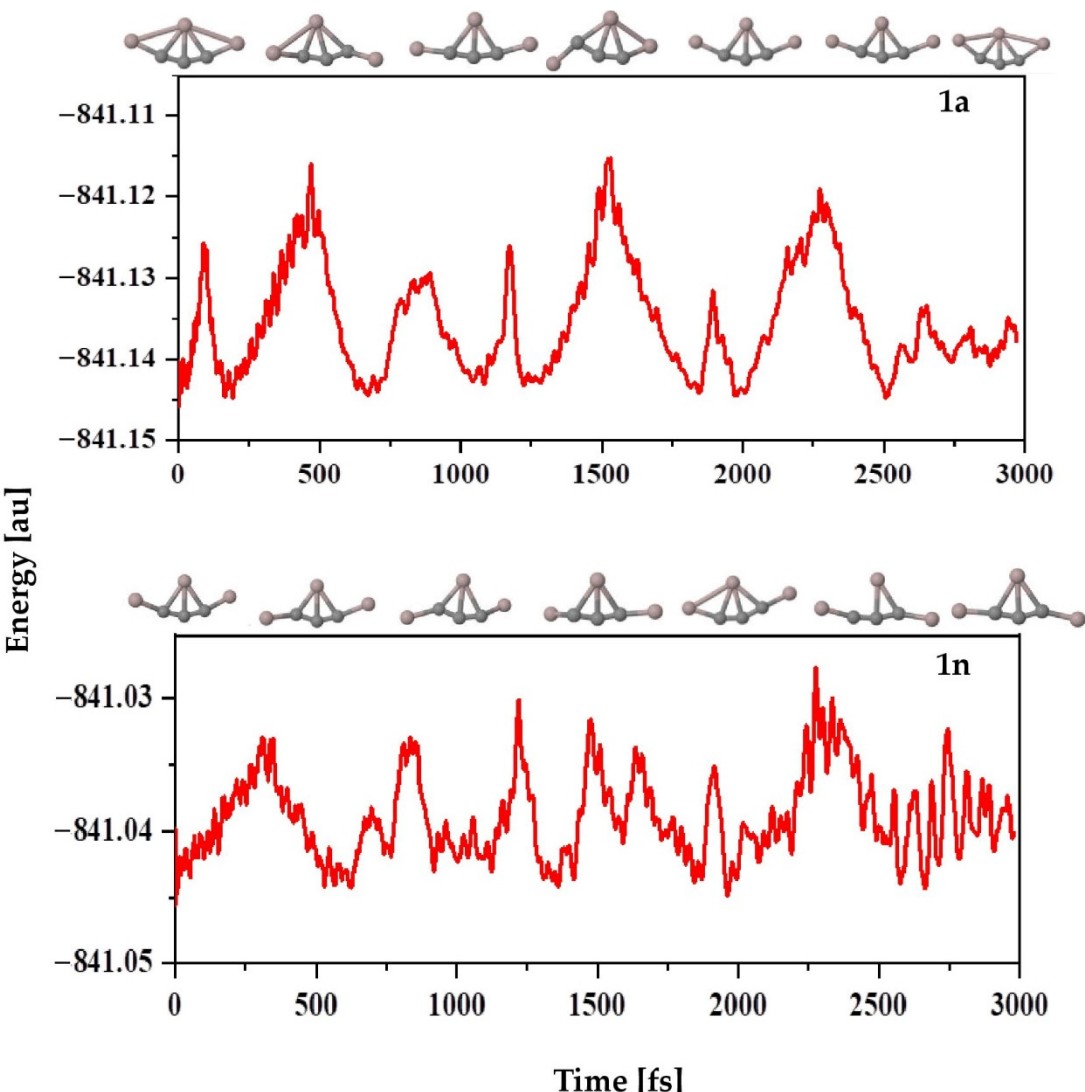

**Figure 6.** Time evolution of the total energy of isomers **1a** (**top**) and **1n** (**bottom**) calculated at the PBE0-D3/def2-TZVP level.

### 4. Conclusions

Using density functional theory, various isomers of the $Al_3C_3^{-/0}$ system were explored first by chemical intuition, and other possible isomers are then identified with the help of the in-house Python code. The global minimum isomers **1a** and **1n** exhibit five bonds to carbon through tri-coordination. The present work reports for the first time five bonds to carbon through tri-coordination in the $Al_3C_3^{-/0}$ system as observed in isomers **1a** and **1n**, respectively. Isomer **1a** is a planar structure with $C_{2v}$ symmetry, whereas isomer **1n** shows $C_2$ symmetry with the terminal aluminum atoms out of the plane. WBI analysis indicates that the central allenic carbon atom in both the isomers (**1a** and **1n**) forms five bonds through tri-coordination and also obeys the octet rule simultaneously. The BCPs from AIM analysis confirm the presence of bond paths between allenic carbon and its adjacent atoms. The aromatic nature that stabilizes both the isomers **1a** and **1n** is well supported by AdNDP, ELF, and NICS analyses. Both the isomers are kinetically stable as inferred from the ab initio molecular dynamics simulations at 1 atm pressure and 298 K temperature up to 3000 fs of time. The obtained results on the $Al_3C_3^{-/0}$ system via computational calculations may encourage experimentalists to design this new class of molecules in the future.

**Supplementary Materials:** The following supporting information can be downloaded at: https://www.mdpi.com/article/10.3390/chemistry5020076/s1, The optimized geometries of all $Al_3C_3^-$ and $Al_3C_3$ isomers are given in Figures S1 and S2, respectively, NICSs in ppm for the isomers **1a** and **1n** are given in Figure S3, AdNDP bonding patterns with occupation numbers (ONs) for isomers **1a** and **1n** are given in Figures S4 and S5, respectively, total energies (in a.u), Zero-point vibrational energies (ZPVEs; in a.u), ZPVE corrected total energies (E+ZPVE; in a.u), relative energies ($\Delta$E+ZPVE; in kcal mol$^{-1}$), and the number of imaginary frequencies (NImag) of all $Al_3C_3^-$ and $Al_3C_3$ isomers at PBE0-D3/def2-TZVP level are given in Tables S1 and S2, respectively, and Cartesian coordinates of all $Al_3C_3^-$ and $Al_3C_3$ isomers at the PBE0-D3/def2-TZVP level are given in Tables S3 and S4, respectively.

**Author Contributions:** Conceptualization, V.S.T. and K.T.; Investigation, A.H.M., V.S.T. and K.T.; Methodology, A.H.M., V.S.T. and K.T.; Supervision, V.S.T. and K.T.; Visualization, A.H.M. and K.T.; Writing—original draft, A.H.M. and V.S.T.; Writing—review and editing, V.S.T. and K.T. All authors have read and agreed to the published version of the manuscript.

**Funding:** This research received no external funding.

**Institutional Review Board Statement:** Not applicable.

**Informed Consent Statement:** Not applicable.

**Data Availability Statement:** Data are available in the article or Supplementary Materials.

**Acknowledgments:** The computational facility provided at the VIT, Vellore, to carry out this work is gratefully acknowledged.

**Conflicts of Interest:** The authors declare no conflict of interest.

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
