# Peer review of "Five Bonds to Carbon through Tri-Coordination in Al3C3/0"

_chemistry, doi:10.3390/chemistry5020076_

Round 1

Reviewer 2 Report

The paper is well written. Overall, the authors provide a clear and concise summary of their computational findings. The paper reports the theoretical establishment of five bonds to a carbon atom through tri-coordination in the Al3C3anion and neutral Al3C3 molecule. The authors use various quantum chemistry tools to analyze the chemical bonding features of these two isomers and shows their kinetic stability at room temperature using molecular dynamics simulations. However, it would have been helpful if the authors provided more information about the potential applications of these newly discovered isomers, as this would provide context for why their discovery is important.
